# Fire-Related Cues Significantly Promote Seed Germination of Some Salt-Tolerant Species from Non-Fire-Prone Saline-Alkaline Grasslands in Northeast China

**DOI:** 10.3390/plants10122675

**Published:** 2021-12-06

**Authors:** Shaoyang Li, Hongyuan Ma, Mark K. J. Ooi

**Affiliations:** 1Northeast Institute of Geography and Agroecology, Chinese Academy of Sciences, Changchun 130102, China; lishaoyang19@mails.ucas.ac.cn; 2University of Chinese Academy of Sciences, Beijing 100049, China; 3School of Grassland Science, Beijing Forestry University, Beijing 100083, China; 4Centre for Sustainable Ecosystem Solutions, School of Biological Sciences, University of Wollongong, Wollongong 2522, Australia; mark.ooi@unsw.edu.au

**Keywords:** seed germination, fire cues, saline-alkaline grassland, salt-tolerant species, smoke water, heat shock

## Abstract

Seed germination in response to fire-related cues has been widely studied in species from fire-prone ecosystems. However, the germination characteristics of species from non-fire-prone ecosystems, such as the saline-alkaline grassland, where fire occasionally occurs accidentally or is used as a management tool, have been less studied. Here, we investigate the effects of different types of fire cues (i.e., heat and smoke water) and their combined effect on the seed germination of 12 species from the saline-alkaline grassland. The results demonstrated that heat shock significantly increased the germination percentage of *Suaeda glauca* and *Kochia scoparia* var. *sieversiana* seeds. Smoke water significantly increased the germination percentage of *Setaria viridis* and *K. scoparia* seeds. However, compared with single fire cue treatments, the combined treatment neither promoted nor inhibited seed germination significantly in most species. These results suggest that fire cues can be used as germination enhancement tools for vegetation restoration and biodiversity protection of the saline-alkaline grassland.

## 1. Introduction

Fire is a fundamental ecological process that profoundly affects many terrestrial ecosystems, including forest and grassland ecosystems [1,2,3]. Fire affects the distribution of global biomes and acts as a natural selection force to drive the evolution of species [4,5]. In the Mediterranean climate zone, savannas, and coniferous forests in the northern hemisphere, fire has become an important ecological factor to maintain the stability of these ecosystems [6].

Fire can affect seed dispersal, germination, seedling establishment, flowering, and other stages of plant growth and development [7]. Seed germination is an important life-history stage, and it is also the main route for vegetation restoration following fire [8]. The immediately post-fire habitat is an advantageous environment for seed germination and seedling establishment, with low competition and high available resources [9].

In addition to creating a favorable external environment for seed germination, the vegetation combustion process produces smoke, heat, and a series of fire cues associated with smoke that play critical roles in promoting seed germination [10,11,12,13]. Heat shock can trigger germination by destroying the thick impermeable seed coat, allowing for water uptake [14]. Plant-derived smoke contains karrikins (KARs), a family of compounds produced by wildfires known to stimulate the germination of dormant seeds of at least 1200 Australian species [15,16,17,18]. Although seed germination in response to fire cues has been extensively studied in fire-prone ecosystems, their effects on seed germination of species from non-fire-prone ecosystems are poorly understood [19,20].

The Songnen plain (121°27′–128°12′ E, 43°36′–49°45′ N) experiences a temperate semi-humid and semi-arid continental monsoon climate [21]. The dry and windy weather that prevails in winter and spring increases the risk of natural fire [22]. As a typical interlaced farming–pastoral zone, farmers commonly burn straws before seeds are sown each spring [23,24]. The germination stimulants in smoke are volatile and water-soluble and can influence the surrounding habitats [25]. The Songnen plain is one of the world’s major saline-alkali areas and is typically ecologically fragile [26]. Salt-alkali stress can inhibit seed germination and seedling growth [27]. Grassland degradation due to climate change and human activities has become a serious environmental problem in the Songnen plain [28]. Therefore, salt-tolerant species, such as *Leymus chinensis* (Trin.) Tzvel and *Suaeda glauca* (Bunge) Bunge, play an important role in the ecological restoration and the sustainable development of animal husbandry [29,30]. Fire cues, especially smoke, provide opportunities for the development of germination tools to promote seed germination and seedling growth in ecological restoration efforts [15,31]. However, less attention has been paid to the effects of fire cues on seed germination of salt-tolerant plants typical of the Songnen plain.

In order to provide theoretical support for the vegetation restoration of degraded habitat and sustainable development of animal husbandry, we selected 12 representative salt-tolerant species in the Songnen saline-alkaline grassland to explore the response of seed germination to different fire cues, alone and in combination, to address the following two questions: (1) Does seed germination of species from Songnen non-fire-prone saline-alkaline grassland respond to fire cues? (2) Is there a combined effect between smoke water and heat shock on specific species?

## 2. Results

### 2.1. Germination Response to Different SW Concentrations

Germination responses to different smoke water (SW) concentrations greatly differed among species (Figure 1). Germination percentages of *Chloris virgata* Sw., *Suaeda corniculate* (C. A. Mey.) Bunge, *Lepidium densiflorum* Schrader, and *Plantago depressa* Willd. were higher than 90% (Figure 1D,F,I,J), while the germination percentages of *Setaria viridis* (L.) Beauv, *Setaria pumila* (Poiret) Roemer & Schultes, *Lespedeza bicolor* Turcz., and *Hibiscus trionum* L. were less than 10% (Figure 1B,C,K,L). Of the 12 tested species, the germination percentages of 11 species did not differ significantly among different SW concentrations (*p* > 0.05, Figure 1). Only the germination of *Kochia scoparia* var. *sieversiana* (Pall.) Ulbr.ex Aschers.et Graebn significantly increased, reaching a maximum germination percentage of 86.75% at the SW concentration of 1%, which was 54.25% higher than the control (df = 5, χ^2^ = 150.29, *p <* 0.001, Figure 1G).

The mean germination time (MGT) of five species differed significantly among different SW concentrations (*p* < 0.05, Figure 2), while the MGT of *L*. *chinensis* and *Suaeda glauca* (Bunge) Bunge was not significantly different from the controls (df = 5, F = 10.03, *p* = 0.074; df = 5, F = 10.255, *p* = 0.068, Figure 2A,C). For *C. virgata* and *L. densiflorum*, the MGT was significantly longer at 3% SW than at lower concentrations (Figure 2B,F). The MGT of *C. virgata* and *P. depressa* increased with the increasing SW concentration (Figure 2B,G). Compared to controls, the MGT of *S. corniculata* differed significantly among different SW concentrations (df = 5, F = 4.732, *p* < 0.01), and the shortest MGT reached 3.02 days at the SW concentration of 1.0% (Figure 2D). The MGT of *K. scoparia* under control treatment was significantly shorter compared to those under other SW concentrations, as the germination percentage was low and the germination was concentrated in the first five days (Figure 2E).

The root length and shoot length responses to different concentrations of SW varied among species (Table 1). The root lengths of *P. depressa* first increased and then decreased with the increasing concentration of SW, reaching a maximum at 1% SW concentrations. The root length of *S. glauca* did not significantly differ among different SW concentrations, while the root length of other species was significantly inhibited by 3% SW. Compared with the root length responses to different SW concentrations, a high SW concentration only inhibited the shoot length of *C. virgata*, *S. corniculate*, and *K. scoparia*.

### 2.2. Germination Response to Heat, SW, and Their Combined Effects

Overall, the germination percentages of four species were significantly affected by heat, SW, and their combined treatments (*p* < 0.05, Figure 3). Compared to the controls, SW significantly increased the germination of *S. viridis* and *K. scoparia* by 25% and 46.44%, respectively (Figure 3B,G). Heat shock significantly increased the germination percentages of *S. glauca* and *K. scoparia* by 33.92% and 22.5%, respectively, while it inhibited the germination of *C. virgata*, which decreased by 13.91% (Figure 3E,G,D). The combined effects of SW and heat shock significantly promoted the germination of *S. viridis*, *S. glauca*, and *K. scoparia* by 22%, 30.92%, and 59.5%, respectively, while it inhibited the germination of *C. virgata*, which decreased by 12.92% (Figure 3B,E,G,D). The germination percentage of *C. virgata*, treated with both heat shock and SW, was significantly lower than those treated with SW alone, which indicated that their combined effect was negative (Figure 3D). However, compared with a single fire cue, the combined effect had no significant positive or negative effect on other species.

Overall, the MGT of six species was significantly affected by heat, SW, and their combined treatments (*p* < 0.05, Figure 4). The MGT of *L. chinensis* and *S. viridis* was not significantly changed by fire cues (df = 3, F = 2.677, *p* = 0.444; df = 3, F = 0.214, *p* = 0.885, Figure 4A,B). The MGT of *S. corniculata* reached a minimum value of 3.02 days under SW treatment, which was significantly shorter compared to the controls (Figure 4E). Heat shock significantly prolonged the MGT of *C. virgata* and *P. depressa* by 0.49 and 1.00 days, respectively, and shortened the MGT of *S. glauca* and *L. densiflorum* by 2.00 and 1.99 days, respectively (Figure 4C,H,D,G). The combined treatment of SW and heat shock significantly shortened the MGT of *S. glauca* and *L. densiflorum* while prolonging the MGT of *C. virgata*, *K. scoparia*, and *P. depressa* (Figure 4D,G,C,F,H). Because the germination percentage of *K. scoparia* was very low under the control treatment, all fire cue treatments significantly increased the germination of *K. scoparia* but also prolonged the germination time.

## 3. Discussion

Plant-derived smoke has been demonstrated to act as a germination cue in a large number of species [8,18,32]. The germination stimulants in smoke, such as karrikinolide (KAR_1_) and glyceronitrile (SP_1_), are responsible for promoting seed germination in some plants from fire-prone or non-fire-prone ecosystems [15,16,33]. Çatav et al. [34] found that the germination percentage of *Satureja thymbra* from the Mediterranean region increased from 57% to 86.5% after treatment with 50% SW [34]. Our results show that SW promoted the germination of *K. scoparia*, *S. viridis*, and other representative species in the non-fire-prone saline-alkaline grassland. In another study on a non-fire-prone ecosystem, Daws et al. [35] found that seed germination of eight weed species from non-fire-prone ecosystems significantly improved after KAR_1_ treatment, indicating that KAR_1_ has wide applicability as a germination stimulant. However, most seeds with physiological dormancy respond to smoke cues only after a period of after-ripening. In this study, under the same treatment, the germination of *S. viridis* in the combined heat shock and SW experiment was higher than that in the SW gradient experiment. The reason for this result was that *S. viridis* seeds underwent a month of after-ripening at room temperature after the end of the SW gradient experiment, which led to a decrease in the physiological dormancy (PD) level. Notably, smoke contains substances that not only stimulate germination but also inhibit it [36]. The composition of plant-derived smoke is complex, and there are still some unknown germination stimuli in smoke cues [37]. Therefore, more experiments on the effects of smoke on seed germination are needed in the future, especially in selecting representative species from different habitats, which will help to identify new specific chemicals in smoke that affect seed germination.

Heat shock breaks the physical dormancy of seeds by destroying the thick impermeable layer of the seed coat to promote seed germination. It has been reported that the physical dormancy of two legumes (*Harpalyce* sp. and *Mimosa leiocephala*) was broken by heat shock, promoting seed germination [38,39,40]. However, in this study, we found that heat shock significantly promoted seed germination of *S. glauca* and *K. scoparia*, which are representative plants in saline-alkaline grassland with PD. This means that heat shock is not the exclusive cue for physically breaking dormancy in dormant seeds. Previous studies have shown that some species with PD also respond to heat shock by germination, e.g., *Drosophyllum lusitanicum*, *Darwinia diminuta*, and *Darwinia fascicularis* [41,42]. In addition to PD, heat shock can also promote the germination of seeds with morphophysiological dormancy (MPD), e.g., *Anigozanthos flavidus* [43].

A high concentration of SW inhibits seed germination and seedling growth. We found that high concentrations of smoke increased the MGT and inhibited the seedling growth of some species. The results are consistent with Chou et al. [44] and Van Staden et al. [45], who showed that high concentrations of SW inhibit germination and seedling growth. As a representative inhibitor, trimethylbutenolide (TMB) has been proven to have the ability to inhibit seed germination in smoke [36]. Therefore, the inhibitory effects of high concentrations of SW on seed germination are probably caused by the influence of germination-inhibiting substances that override the effects of KARs [44]. Temperature and the duration of exposure are two important heat-shock-associated variables that affect seed germination. In this study, we found that heat shock at 80 °C for 10 min significantly inhibited the germination of *C. virgata* and prolonged the MGT of *C. virgata* and *P. depressa*. This indicates that the heat shock intensity exceeded the tolerance threshold of these species. Previous studies have shown that the germination of *Melica ciliate* reached a maximum under heat shock at 110 °C for 5 min, but it was completely inhibited at 150 °C for 10 min [13]. To better simulate the effects of fire cues on seed germination, it is necessary to investigate the fire regime of different ecosystems, such as forests and grasslands, which can vary greatly.

The combined treatment of heat shock and SW may show positive, negative, or no effect on seed germination [43,46]. The effects of combined treatment may be related to many factors, such as seed dormancy type and fire cue intensity. In this study, the combined treatment of heat shock and SW significantly inhibited the germination of *C. virgata*, while there was no significant positive or negative effect on other species. This indicates that the combined intensity of SW and heat shock exceeds the seed germination requirements of *C. virgata*. For species with combinational dormancy, heat shock breaks the physical dormancy barriers of seeds, and subsequent addition of SW could promote the release of seeds from their physiological dormancy to achieve a higher germination percentage [42]. Seeds with physical dormancy are usually unable to respond to smoke due to the impermeability of the seed coat.

As a typical ecological fragile region, saline-alkaline stress is the main limiting factor of seed germination in the Songnen saline-alkaline grassland [27,47]. The soil seed bank is an essential part of degraded vegetation dynamics, which directly impacts the resilience of the ecosystem [48]. In nature, germination-promoting substances of smoke penetrate into soil with rain and stimulate the germination of seeds in the soil seed bank [49]. Therefore, the responses of seed germination of salt-tolerant plants to fire cues, such as *S. glauca*, *K. scoparia*, suggest that fire cues could be used as germination enhancement tools to help vegetation restoration in degraded ecosystems.

## 4. Materials and Methods

### 4.1. Seed Collection

Mature seeds were collected from at least 20 individuals of each of the 12 species in Songnen saline-alkaline grassland. All individuals were located in the same location, with each individual approximately 0.5 m from any other study plant. Seeds were dry stored in paper bags at room temperature (~22 °C) until the experiments commenced in November 2020 (Table 2).

### 4.2. Germination Experiment

One kilogram of dried plant material from native vegetation of Western Australia (i.e., Proteaceae, Fabaceae, Rutaceae) was fully burnt to produce smoke, which was subsequently pumped through 5 L of distilled water for 2.5 h to produce a 100%, fully saturated concentration of smoke water [50]. Deionized water was used to dilute the saturated SW to 0.5%, 1%, 1.5%, 2%, and 3% concentrations, representing 5 different treatment concentrations of SW [50,51]. Deionized water (0% SW) was used as the control. For the germination experiment, seeds were placed in the germination medium containing 0.7% agar and used with the above-mentioned SW concentrations.

In the heat shock treatments, seeds were put in an oven and heat-shocked at 80 °C for 10 min [52]. Heat-shock-treated seeds were placed in Petri dishes (9 cm in diameter) with 0.7% agar medium. In the experiments testing the combined effects of heat shock and SW, seeds were first put in an oven and heat-shocked at 80 °C for 10 min and then placed in the 0.7% agar medium containing 1% SW concentration for germination experiments.

For all experiments, 25 seeds were placed in each Petri dish, and each treatment was repeated four times. Petri dishes were kept in an incubator (Harbin, China) at 16/28 °C with a 12/12 h dark/light diurnal cycle [47,53]. Germination was checked once a day, and the experiment was stopped when no germination was observed for three consecutive days. On the third day after the seed germination test, five seedlings were taken from each Petri dish to measure the lengths of roots and shoots. The tested species that had germination percentage < 25% were excluded from root length and shoot length analysis.

### 4.3. Statistical Analysis

Germination percentage and mean germination time (MGT) were calculated using the following formulas [54,55]:$$\mathrm{Germination}\;\mathrm{percentage}=\frac{n}{N}\;\times 100\%;\;\mathrm{MGT}=\frac{\sum it}{\sum i}$$
where *n* is the number of final germinated seeds, *N* is the total number of seeds, *i* is the number of seeds newly germinated on the day *t*, and *t* is the number of days counted from the beginning of the germination experiment. The tested species that had germination percentage <25% were excluded from MGT, root length, and shoot length analyses.

We used generalized linear models (GLMs) with binomial error structure and the log link function to evaluate the effects of different treatments on germination. The linear model was used to analyze the effects of different treatments on MGT, root length, and shoot length. Tukey’s test was used for multiple mean comparisons. All statistical analyses and figures were carried out in R 3.6.3.

## 5. Conclusions

In conclusion, our results show that fire cues significantly promoted the germination of some salt-tolerant species, e.g., *S. viridis*, *K. scoparia*, and *S. glauca*, which are representative plants in the saline-alkaline grassland. The seed germination responses to different types and intensities of fire cues varied among species. Since salt-tolerant plants are the key species involved in vegetation restoration in degraded saline-alkali grassland, understanding the response of seed germination to fire cues is of great significance for future restoration of degraded grassland and sustainable development of animal husbandry in these ecologically fragile areas.

## Figures and Tables

**Figure 1 plants-10-02675-f001:**
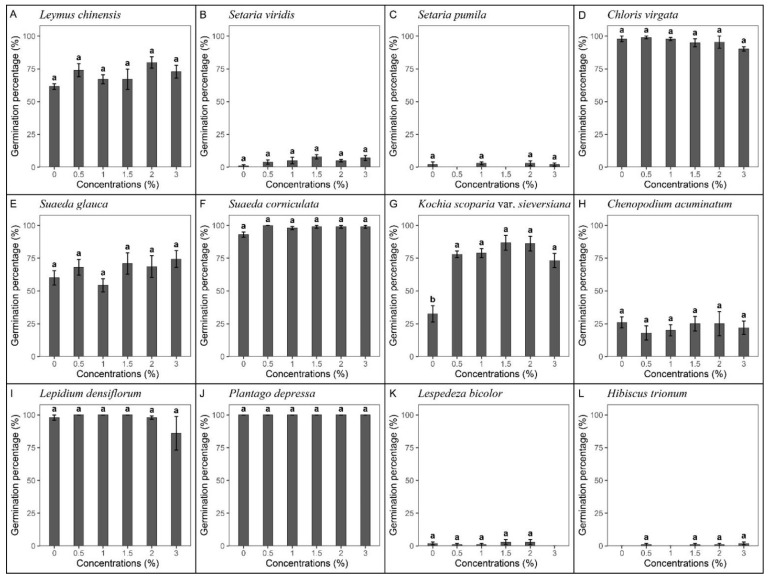
Germination percentage of 12 species in response to different concentrations of smoke water at 16/28 °C with a 12/12 h dark/light diurnal cycle. The numbers (0, 0.5, 1, 1.5, 2, 3) represent the concentration degrees of smoke-water dilution. (**A**) *Leymus chinensis*, (**B**) *Setaria viridis,* (**C**) *Setaria pumila*, (**D**) *Chloris virgata*, (**E**) *Suaeda glauca*, (**F**) *Suaeda corniculate,* (**G**) *Kochia scoparia* var. *sieversiana*, (**H**) *Chenopodium acuminatum*, (**I**) *Lepidium densiflorum*, (**J**) *Plantago depressa*, (**K**) *Lespedeza bicolor*, and (**L**) *Hibiscus trionum.* Bars show means ± SE (n = 5). Different letters mean significant differences in germination percentage among the concentrations (*p* < 0.05).

**Figure 2 plants-10-02675-f002:**
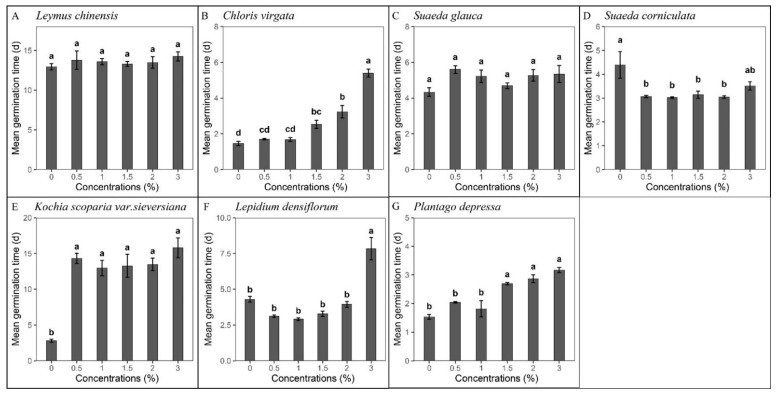
MGT of 7 species in response to different concentrations of smoke water at 16/28 °C with a 12/12 h dark/light diurnal cycle. Species with germination percentages lower than 25% were not analyzed. The numbers (0, 0.5, 1, 1.5, 2, 3) represent the concentration degrees of smoke-water dilution. (**A**) *Leymus chinensis*, (**B**) *Chloris virgata*, (**C**) *Suaeda glauca*, (**D**) *Suaeda corniculate,* (**E**) *Kochia scoparia* var. *sieversiana*, (**F**) *Lepidium densiflorum*, and (**G**) *Plantago depressa*. Bars show means ± SE (n = 5). Different letters indicate significant differences in MGT among the concentrations (*p* < 0.05).

**Figure 3 plants-10-02675-f003:**
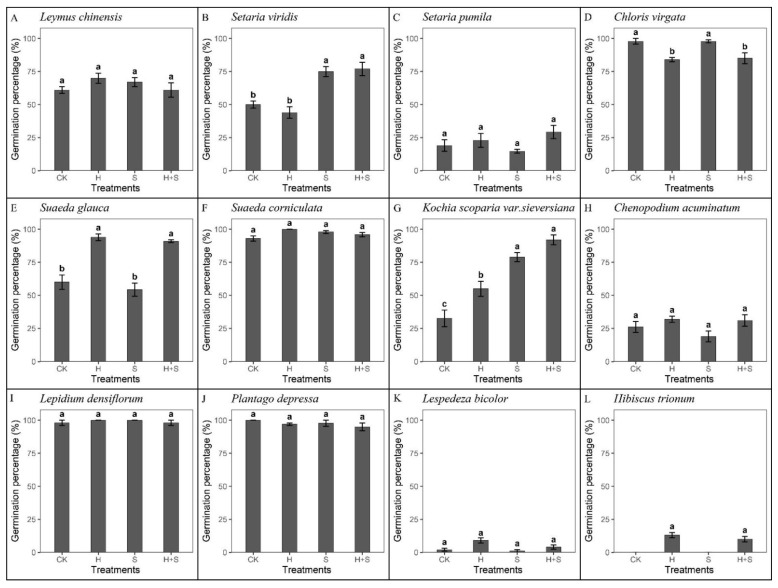
Germination percentage of 12 species in response to different fire cues at 16/28 °C with a 12/12 h dark/light diurnal cycle. CK, control; H, heat shock at 80 °C for 10 min; S, 1% smoke water; H + S, the combined treatment of heat shock and smoke water. (**A**) *Leymus chinensis*, (**B**) *Setaria viridis,* (**C**) *Setaria pumila*, (**D**) *Chloris virgata*, (**E**) *Suaeda glauca*, (**F**) *Suaeda corniculate,* (**G**) *Kochia scoparia* var. *sieversiana*, (**H**) *Chenopodium acuminatum*, (**I**) *Lepidium densiflorum*, (**J**) *Plantago depressa*, (**K**) *Lespedeza bicolor*, and (**L**) *Hibiscus trionum.* Bars show means ± SE (n = 5). Different letters mean significant differences in germination percentage among fire cues (*p* < 0.05).

**Figure 4 plants-10-02675-f004:**
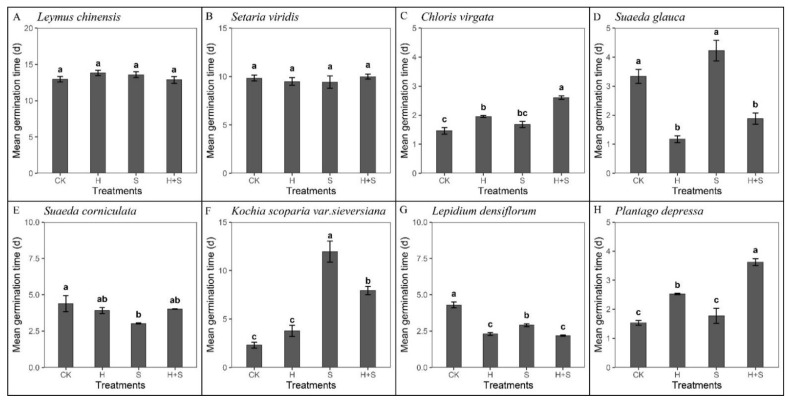
MGT of 8 species in response to different fire cues at 16/28 °C with a 12/12 h dark/light diurnal cycle. Species with germination percentages lower than 25% were not analyzed. CK, control; H, heat shock at 80 °C for 10 min; S, 1% smoke water; H + S, the combined treatment of heat shock and smoke water. (**A**) *Leymus chinensis*, (**B**) *Setaria viridis,* (**C**) *Chloris virgata*, (**D**) *Suaeda glauca*, (**E**) *Suaeda corniculate,* (**F**) *Kochia scoparia* var. *sieversiana*, (**G**) *Lepidium densiflorum*, and *(***H**) *Plantago depressa.* Bars show means ± SE (n = 5). Different letters mean significant differences in MGT among fire cues (*p* < 0.05).

**Table 1 plants-10-02675-t001:** Root and shoot length (cm) in response to different concentrations of smoke water.

Species	Length	0	0.5%	1%	1.5%	2%	3%
*L. chinensis*	root	5.70 ± 0.24 a	5.21 ± 0.21 ab	4.37 ± 0.26 b	5.21 ± 0.19 ab	5.07 ± 0.28 ab	4.32 ± 0.51 b
	shoot	5.13 ± 0.13 a	4.90 ± 0.12 a	5.09 ± 0.11 a	5.19 ± 0.13 a	5.32 ± 0.18 a	5.00 ± 0.24 a
*C. virgata*	root	7.01 ± 0.41 a	5.72 ± 0.32 a	4.16 ± 0.43 b	2.89 ± 0.28 bc	2.38 ± 0.30 c	2.81 ± 0.50 bc
	shoot	2.48 ± 0.08 a	2.48 ± 0.22 a	1.91 ± 0.10 b	1.85 ± 0.08 b	1.95 ± 0.11 ab	1.75 ± 0.33 b
*S. glauca*	root	3.21 ± 0.28 a	2.66 ± 0.26 a	2.73 ± 0.32 a	2.47 ± 0.25 a	3.43 ± 0.30 a	2.74 ± 0.25 a
	shoot	3.05 ± 0.25 a	3.20 ± 0.20 a	3.17 ± 0.17 a	3.31 ± 0.27 a	3.29 ± 0.15 a	3.10 ± 0.16 a
*S. corniculata*	root	2.02 ± 0.09 a	0.99 ± 0.08 b	0.68 ± 0.10 bc	0.52 ± 0.11 cd	0.24 ± 0.05 d	0.33 ± 0.05 cd
	shoot	1.05 ± 0.03 ab	1.09 ± 0.02 ab	0.99 ± 0.04 ab	1.50 ± 0.35 a	0.81 ± 0.06 b	0.59 ± 0.05 b
*K. scoparia*	root	4.81 ± 0.62 a	4.59 ± 0.47 a	4.42 ± 0.34 a	3.33 ± 0.43 ab	2.78 ± 0.39 b	1.10 ± 0.32 c
	shoot	1.87 ± 0.19 a	1.23 ± 0.07 b	1.22 ± 0.05 b	1.07 ± 0.06 bc	1.34 ± 0.07 b	0.83 ± 0.06 c
*L. densiflorum*	root	4.18 ± 0.16 a	4.03 ± 0.18 a	2.23 ± 0.19 b	0.33 ± 0.21 c	1.01 ± 0.21 c	0.61 ± 0.07 c
	shoot	0.51 ± 0.01 a	0.85 ± 0.24 a	0.67 ± 0.24 a	0.33 ± 0.02 a	0.43 ± 0.02 a	0.70 ± 0.23 a
*P. depressa*	root	3.51 ± 0.22 ab	4.05 ± 0.17 a	4.28 ± 0.22 a	2.98 ± 0.31 bc	2.23 ± 1.31 c	0.34 ± 0.08 d
	shoot	0.70 ± 0.03 a	0.67 ± 0.02 a	0.93 ± 0.20 a	0.68 ± 0.04 a	0.87 ± 0.27 a	0.46 ± 0.02 a

On the third day after the seed germination test, five seedlings were taken from each Petri dish to measure the lengths of roots and shoots. Species with germination percentages lower than 25% were not analyzed. Numerical values show means ± SE (n = 5). Different letters mean significant differences in the lengths of roots and shoots among the concentrations (*p* < 0.05).

**Table 2 plants-10-02675-t002:** Information on the test species.

Species	Family	Collection time	GD (days)
*L. chinensis*	Poaceae	July 2020	28
*S. viridis*	Poaceae	November 2020	26
*S. pumila*	Poaceae	September 2020	26
*C. virgata*	Poaceae	September 2020	8
*K. scoparia*	Chenopodiaceae	October 2020	25
*S. glauca*	Chenopodiaceae	October 2020	10
*S. corniculata*	Chenopodiaceae	September 2020	10
*C. acuminatum*	Chenopodiaceae	September 2020	10
*L. densiflorum*	Brassicaceae	July 2020	10
*P. depressa*	Plantaginaceae	July 2020	10
*L. bicolor*	Fabaceae	October 2020	9
*H. trionum*	Malvaceae	October 2020	9

GD: duration of germination experiment from start to end.

## Data Availability

The data presented in this study are available in the article.

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
