# Peer review of "Fire-Related Cues Significantly Promote Seed Germination of Some Salt-Tolerant Species from Non-Fire-Prone Saline-Alkaline Grasslands in Northeast China"

_plants, 2021, doi:10.3390/plants10122675_

Round 1

Reviewer 1 Report

An interesting paper that needs some serious improvement, especially in the Methods and Material section, before accepted for publication.

Specific comments

Line 226. Authors should give some more information for the amount of seeds collected per species, as well as for the individuals selected (e.g. distance between them, at the same or different locations, etc.).

Line 233. Authors write “One kilogram of dried plant material…”. Here, they should explain where they found this material. Also, they should make clear if this material concerns each species of the 12 species studied.

Line 236. Authors should justify why they selected these specific treatment concentrations of SW. References are needed.

Authors should give all the information about the time of the experiments, their duration, the dates etc.

Line 247. Authors should justify why they selected this cycle (16/28 °C with a 12/12 h dark/light diurnal cycle). References are needed.

Line 245-246.  ISTA relevant recommendations?

Line 249. Authors should report the time for“measuring the length of roots and shoots”.

Line 253. Authors should cite a few recent relevant references for using the term “mean germination time (MGT)”, e.g. 1) Regeneration Ecology of the Rare Plant Species Verbascum dingleri: Implications for Species Conservation. Sustainability 2019, 11, 3305. https://doi.org/10.3390/su11123305. 2) Intraspecific differences in the response to drying of Quercus ithaburensis acorns, Plant Biosystems 2017, 151:5, 878 886, DOI: 10.1080/11263504.2016.1219415

Line 72. In the beginning of the Results section, subsection 2.1. Germination Response to Different SW Concentrations, it is better the authors to start the first Result paragraph with reporting that: “It was observed a great differentiation of species germination responses to different SW concentrations; most studied species presented a high germination percentage in all SW treatments (Chloris virgata Sw., Suaeda corniculate (C. A. Mey.) Bunge, Lepidium densiflorum Schrader, and Plantago depressa.......) while some others exhibited a very low germination percentage (less than 10%), ………….”

Because this is a basic finding of the study.

And then to comment the differentiation among the different  SW concentrations within the studied species.

Table.1 Authors should mention in the Table title (as well as in the text) the time of shoot and root measurements.

Line 201, in the Discussion section. Authors should discuss and analyze their finding “that high concentrations of smoke inhibited the seedling growth of some species”, while some others not.

They should also justify this ecological behavior of the studied species, and report any conclusion for management implications, that is missing in the manuscript.

Reviewer 2 Report

The paper is nicely written. The results are clearly presented and discussed. I have no objections concerning this paper. However I have just few remarks and some minor recommendations.

When you prepare SW from 1kg of dried plant material in my opinion the more detailed characteristic of this plant material is needed. I can imagine that from different plant composition the obtained smoke and SW would be composed from different chemicals, which could exert different reaction of seeds.

For the measurement of roots and shoots length you should precise the time/day of experiment. For eg. DAS (day after sowing) or DAG (day after germination).

Some minor remarks:

The list of the species should be introduced in the introduction or in the beginning of the results.

All abbreviation should be explained when there are used for the first time (MGT in line 89 and SW in line 75. I advise you to add information on figures that the used concentrations are dilution of SW. It could be in figure caption. Now there is just information “concentration %”.
